# Phylogenetic, Evolutionary and Structural Analysis of Canine Parvovirus (CPV-2) Antigenic Variants Circulating in Colombia

**DOI:** 10.3390/v12050500

**Published:** 2020-04-30

**Authors:** Sebastián Giraldo-Ramirez, Santiago Rendon-Marin, Julián Ruiz-Saenz

**Affiliations:** Grupo de Investigación en Ciencias Animales—GRICA, Facultad de Medicina Veterinaria y Zootecnia, Universidad Cooperativa de Colombia, Sede Bucaramanga 680005, Colombia; sebastian.giraldor2@udea.edu.co (S.G.-R.); santiago.rendonm@udea.edu.co (S.R.-M.)

**Keywords:** antigenicity, sequencing, virus

## Abstract

Canine parvovirus (CPV-2) is the causative agent of haemorrhagic gastroenteritis in canids. Three antigenic variants—CPV-2a, CPV-2b and CPV-2c—have been described, which are determined by variations at residue 426 of the VP2 capsid protein. In Colombia, the CPV-2a and CPV-2b antigenic variants have previously been reported through partial VP2 sequencing. Mutations at residues Asn428Asp and Ala514Ser of variant CPV-2a were detected, implying the appearance of a possible new CPV-2a variant in Colombia. The purpose of the present study was to characterise the full VP2 capsid protein in samples from Antioquia, Colombia. We conducted a cross-sectional study with 56 stool samples from dogs showing clinical symptoms of parvoviral disease. Following DNA extraction from the samples, VP2 amplification was performed using PCR and positive samples were sequenced. Sequence and phylogenetic analyses were performed by comparison with the VP2 gene sequences of the different CPV-2 worldwide. VP2 was amplified in 51.8% of the analysed samples. Sequencing and sequence alignment showed that 93.1% of the amplified samples belonged to the new CPV-2a antigenic variant previously. Analysing the amino acid sequences revealed that all CPV-2a contain Ala297Asn mutations, which are related to the South America I clade, and the Ala514Ser mutation, which allows characterization as a new CPV-2a sub-variant. The Colombian CPV-2b variant presented Phe267Tyr, Tyr324Ile and Thr440Ala, which are related to the Asia-I clade variants. The CPV-2c was not detected in the samples. In conclusion, two antigenic CPV-2 variants of two geographically distant origins are circulating in Colombia. It is crucial to continue characterising CPV-2 to elucidate the molecular dynamics of the virus and to detect new CPV-2 variants that could be becoming highly prevalent in the region.

## 1. Introduction

One of the main infectious agents that affect the canine population is the canine parvovirus type 2 (CPV-2), which causes acute haemorrhagic diarrhoea, primarily in puppies [1]. CPV-2 is a single-stranded DNA virus that belongs to the genus *Protoparvovirus* [2], with a genome of approximately 5200 nucleotides that contain two open reading frames (ORFs). The ORF 3′ codes for non-structural proteins NS1 and NS2 are important in controlling viral replication and assembly. The ORF 5′ codes for structural proteins are termed viral protein 1 (VP1) and 2 (VP2). VP2 is the major component of the viral capsid, which has an icosahedral structure of 60 subunits with a T = 1 symmetry, comprising approximately 4–5 copies of VP1 and 54–55 copies of VP2 [3].

VP2 subunits interact with transferrin receptors (TfRs) present on the outer surface of the cell membrane [4]. Subsequently, the absorption to the host cell is facilitated by clathrin-mediated endocytosis [5]. The VP2 capsid protein plays a crucial role because its mutations determine the antigenic changes that originate in the different antigenic CPV-2 variants [4]. 

It is believed that CPV-2 is derived from the feline panleukopenia virus (FPLV) [6], where specific mutations at Lys80Arg, Lys93Asn, Val103Ala, Asp323Asn, Asn564Ser and Ala568Gly capsid protein VP2 residues facilitated a change of host, thereby allowing the virus to infect canines and losing the ability to infect felines [7,8]. These mutations originated in the CPV-2 variant, firstly reported in the 1970s, and spread to countries in Europe, America, Asia and Oceania by 1978 [6].

By 1982, CPV-2 was replaced by a variant of the virus that genetically and antigenically differed [9]. This variant was called CPV-2a and differs from CPV-2 in 6 amino acids: Met87Leu, Ile101Thr, Ser297Ala, Ala300Gly, Asp305Tyr and Val555Ile residues [10]. 

Residue 426 of VP2 is located in the outermost part of the threefold axis, where 3 VP2 subunits converge. It is the site of greatest antigenicity of the virus [11]; therefore, the amino acid variation causing the antigenic changes that led to the origin of the CPV-2b antigenic variants (Asn426Asp) was reported in 1984 [10] and that for CPV-2c (Asp426Glu) was reported in Italy in 2001 [12]. Unlike CPV-2, CPV-2a, CPV-2b and CPV-2c variants infect canines as well as regaining the ability to infect felines and other wild carnivores [8,13]

In 2017, the antigenic CPV-2a and CPV-2b variants were reported via the study of a partial region of VP2 in Colombia [14]. The CPV-2c variant was not detected in Colombia, despite being the most prevalent antigenic variant in South America [15] and despite Colombia sharing borders with countries such as Peru, Ecuador and Brazil, where the variant has previously been reported [16]. Additionally, two mutations were observed in the Colombian CPV-2a variants (Asn428Asp and Ala514Ser) that suggest the emergence of a new CPV-2a variant unique in Colombia [14].

The aim of the present study was to characterise the coding region of the entire VP2 capsid protein of the circulating parvovirus variants in Antioquia (north-western Colombia) for determining the total amino acid variations in VP2 and to obtain information regarding the molecular evolution of the CPV-2 in Colombia.

## 2. Materials and Methods

### 2.1. Patient Selection and Sampling

A cross-sectional study was conducted with a convenience sampling of canine patients attending different veterinary clinics of the department of Antioquia and reporting clinical symptoms compatible with canine parvovirus, such as: haemorrhagic diarrhoea, dehydration, vomiting, loss of appetite and weakness. The samples were collected from February 2018 to March 2019. Gender, breed, age, and vaccination status of each patient was registered. No previous confirmation nor positive faecal antigen ELISA were required for any patient. Diagnosis was based on typical clinical signs. Faecal samples of each animal were collected (approximately 5 g) and stored under freezing conditions (−80 °C) until further use; prior authorisation was obtained from the owner of the animals that met the inclusion criteria. This study was approved by the Bioethics Committee of the Universidad Cooperativa de Colombia (project INV1473–bioethics Acta 002-2016). An informed consent was obtained from all owners.

### 2.2. DNA Extraction and Quantification

Viral DNA extraction from the collected stool samples was performed using QIAamp DNA fast stool mini kit (Qiagen^®^, Hilden, Germany), in accordance with the manufacturer’s instructions. DNA obtained was quantified using 1 μL of the product with NanoDrop 2000 (Thermo Fisher Scientific^®^, Waltham, MA, USA).

### 2.3. VP2 Amplification Using PCR

To identify CPV-2-positive samples, PCR amplification of VP2 was conducted using the conventional method. For each PCR reaction, 25 µL of DreamTaqTM PCR Master Mix (2×) (Thermo Fisher Scientific^®^) was used, with 4 µL of the Forward Ext1F primer (5′-ATGAGTGATGGAGCAGTTCA-3′) and 4 µL of the Ext3R Reverse primer (5′-AGGTGCTAGTTGAGATTTTTCATATAC-3′) described by [17] and 17 µL of a mixture of DNA and molecular grade water, reaching an amount of 500 ng DNA for each reaction. Molecular grade water was used as a negative control for amplifications. PCR protocol used was as follows: an initial denaturation cycle at 94 °C for 5 min, 35 denaturation cycles at 94 °C for 30 s, alignment at 50 °C for 45 s, extension at 72 °C for 1 min and a final extension cycle at 72 °C for 5 min.

PCR amplification results were visualised using 1.5% horizontal agarose gel electrophoresis. Gels were stained with the Invitrogen ™ SYBR® Safe DNA Gel Stain (Thermo Fisher Scientific^®^). In each well, 4.2 μL of each sample obtained after amplification and 0.8 μL of the 6× DNA loading buffer were used, and the GeneRuler™ 100-bp DNA Plus Ladder (Thermo Fisher Scientific^®^) was used as a molecular weight marker. Gels were developed in the ultraviolet light Gel Doc™ XR+ imaging system (Bio-Rad, Molecular imager^®^, USA) and were visualised using ImageLab™ software.

### 2.4. Sequencing and Sequence Analysis

Samples positive for VP2 amplification after electrophoresis visualisation were purified and sequenced at Macrogen Inc. (Seoul, Korea) using Forward Ext1F and Reverse Ext 3R primers, along with a set of internal sequencing primers to amplify the entire VP2 [18]—F1: 5′-AGATAGTAATAATACTATGCCATTT-3′, F2: 5′-ACAGGAGAAACACCTGAGAGATTTA-3′, R1: 5′-TGGTTGGTTTCCATGGATA-3′, and R2: 5′-TTTTGAATCCAATCTCCTTCTGGAT-3′. The resulting electropherograms from the sequencing were analysed using Chormas™ v. 2.6 software. Contig generation, resulting from the overlapping of the sequences amplified by the primers, was performed on the SeqMan Pro platform with Lasergene™. After constructing the complete nucleotide sequences for each sample, alignment was performed using the ClustalW method, following which these sequences were compared with the DNA sequences of CPV-2a, CPV-2b and CPV-2c obtained from GenBank. All analyses were performed in the MEGA™ 7.0 software for Windows^®^.

### 2.5. Phylogenetic Analysis

For the phylogenetic analysis, we calculated the best nucleotide substitution model for the dataset generated with the sequences of FPLV, CPV-2, CPV-2a, CPV-2b and CPV-2c obtained from GenBank. Phylogenetic analysis was inferred using distance-based (neighbor-joining) and character based (maximum likelihood—Bayesian) approaches implemented in MEGA™ 7.0 and Mr Bayes™ software for Windows^®^. The nucleotide replacement model selected was Tamura-3 parameter with Gamma distributed rate and invariant sites (T92+G+I) and then the Markov chain Monte Carlo (MCMC) Bayesian analysis. We ran the MCMC searches for 1,000,000 generations. The TRACER™ v1.7.1 software was used to confirm all the parameters generated in the Bayesian analysis. The effective sample size was up to 200. The FigTree v1.4.3 software was used to display the consensus phylogenetic tree generated after Bayesian analysis.

### 2.6. Evolutionary Analysis

The construction of the phylogenetic evolutionary tree required the generation of a dataset containing the sequences obtained from GenBank and the Colombian CPV-2a and CPV-2b samples. The evolutionary rates, time to the most recent common ancestor (tMRCA) and geographic movements of CPV-2 were performed using the BEAST v1.8.4 software package. The phylogenetic evolutionary tree was generated according to the Hasegawa, Kishino and Yano nucleotide substitution model + gamma distribution + invariable sites (HKY + G + I) and a strict molecular clock. The Bayesian stochastic search variable selection was used to determine links between sequences. The length of the Markov chain Monte Carlo chain was 15 million. All parameters generated in the analysis were confirmed by verifying the effective sample size of >200 using the TRACER v1.7.1 software. With TreeAnnotator v1.8.4 software, 10% of the steps (1.5 million burn-in) were eliminated to obtain the tree with the most credible clades. FigTree v1.4.3 software was used to display the generated tree.

### 2.7. Structural Analysis

The VP2 tertiary structure construction and surface analysis was performed using the sequences of sample 1 (CPV-2a) and sample 50 (CPV-2b) as references. Homology modeling of the CPV-2a and CPV-2b sequences under study was generated using MODELLER software based on PDB: 1C8D structure. The three-dimensional models of the VP2 tertiary structure were created using the PyMOL™ software. VP2 surface analysis was performed using the RIVEM (Radial Interpretation of Viral Electron density Maps) software that facilitates the generation of a ‘road map’ of the viral surface and determination of the locations of the amino acids that constitute the analysed structure [19]. 

### 2.8. Analysis of Selection Pressure

The relationship of non-synonymous (dN) to synonymous (dS) substitutions was calculated using ML phylogenetic reconstruction and the general reversible nucleotide substitution model available through the web program Datamonkey. To detect non-neutral selection, the Fast, Unconstrained Bayesian AppRoximation (FUBAR) was implemented in the Datamonkey program. The values dN/dS > 1, dN/dS = 1, and dN/dS <1 were used to define positive selection (adaptive molecular evolution), neutral mutations, and negative selection (purification selection), respectively. Also, we implemented FEL (Fixed Effects Likelihood) which uses a ML approach to infer nonsynonymous (dN) and synonymous (dS) substitution rates on a per-site basis for a given coding alignment and corresponding phylogeny and permits the identification of positive selection at specific sites along particular clades.

### 2.9. Statistical Analyses

A descriptive analysis of the collected data was performed representing the qualitative and quantitative variables in tables and graphs.

## 3. Results

For this study, a total of 56 faecal matter samples were collected from dogs that presented clinical symptoms compatible with canine parvovirus, from different veterinary centres located in the department of Antioquia, Colombia.

PCR amplification of VP2 showed that a total of 29 samples (51.8%) were CPV-2-positive. Of these samples, a total of 41.4% (*n* = 12) belonged to females and 58.6% (*n* = 17) to males. Approximately 55.1% of the positive samples belonged to mixed-breed animals. According to the data provided at the time of sample collection, when visiting the veterinary centres with symptoms compatible with those of parvovirus infection, 20.7% (*n* = 6) of the animals had undergone a complete vaccination schedule as required by age, whereas 44.8% (*n* = 13) did not comply with the vaccination schedule. The remaining 34.5% (*n* = 10) of the animals presented incomplete vaccination schedules (Table 1).

According to age distribution, 15 positive samples (51.7%) belonged to animals aged 1–3 months, which was the most prevalent age group for the virus, whereas only 1 sample (3.4%) belonged to an animal aged >1 year (Figure 1).

### 3.1. Sequence Analysis

An almost full sequence of 1711 nucleotides of the VP2 gene was achieved. Sequencing analysis showed that the antigenic variants present in the study are CPV-2a and CPV-2b. The CPV-2a variant (Asn426) was present in 27 from the positive samples (93.1%) and was the most prevalent variant in the study, whereas the CPV-2b variant (Asp426) was detected in two samples (6.9%). No evidence was obtained regarding the presence of the CPV-2c variant (Glu426) in any sample (Table 1).

Sequence analysis revealed specific mutations with respect to the reference sequences of CPV-2a, CPV-2b and CPV-2c obtained from GenBank. The Colombian CPV-2a variants showed mutations at the amino acid residues Ala297Asn, Tyr324Ile and Ala514Ser, with the latter being reported for the first time in Colombia in 2017 [14]. Regarding the CPV-2b variants in the present study, the amino acid variations detected were Phe267Tyr and Thr440Ala. Similar to the CPV-2a variants, the CPV-2b sequenced from the sampling showed the Tyr324Ile mutation; however, a single Ala514Ser mutation was not detected in these samples (Table 2).

### 3.2. Phylogenetic Analysis

The phylogenetic relationships based on the nucleotide alignment of VP2 Sequences inferred by distance (neighbor joining) and character approaches (maximum likelihood and Bayesian inference) resulted in trees with a similar topology. The phylogenetic tree generated by Bayesian inference included the CPV-2 positive samples representing the antigenic variants detected in the study (CPV-2a and CPV-2b). GenBank accession numbers for Colombian nucleotide sequences are: MT152347 to MT152375. Additionally, these samples exhibited differences in the pairwise distances. Representative samples of FLPV, CPV-2, CPV2a, CPV-2b and CPV-2c from different countries reported in GenBank were used for phylogenetic evolutionary tree construction. The Colombian CPV-2a antigenic variant constituted a monophyletic clade that substantially differs from the European CPV-2a antigenic variants as well as from Uruguayan variants and only shared distribution with an Ecuadorian CPV-2a variant (MG264075). The two samples of the study from the antigenic CPV-2b variant were located within a clade that includes the Uruguayan CPV-2a variants and Asian CPV-2b variants (Figure 2).

In 2017, by using partial VP-2 sequences, the presence of “new” CPV-2 variants in Colombia was shown [14]. To confirm this, we performed maximum likelihood phylogenetic analysis using VP-2 partial sequences, including those that showed the change Ala514Ser in CPV-2a and the sequences belonging to CPV-2b (Appendix A). The analysis reveals that the CPV-2a variants of both studies have strong similar sequences. The CPV-2a sequences from 2017 and current analysis showed minor nucleotide differences; however, they turn out to be synonymous variations that do not represent amino acid changes, and for this reason they are all in the same clade. Regarding the CPV-2b sequences, it can also be seen that they are very similar and are located in a single well differentiated clade. Clade 2b sequences show variations at the amino acid level. CPV-2b 2019 sequences (M50 and M55) are located on a separate branch. This subdivision is due to the fact that only the CPV-2b 2019 sequences present the Thr440Ala mutation, one of the mutations reported in the present work (Appendix A).

### 3.3. Evolutionary Analysis

For the evolutionary analysis, the same sequences of samples used in the phylogenetic analysis and a dataset of the sequences of CPV-2a, CPV-2b and CPV-2c variants from different countries were used. The estimated tMRCA for the phylogenetic evolutionary tree was generated 40 years ago (1979), based on the most recent analysed sequence (2019), with a 95% highest probability density and a range of 38–44 years. The analysis revealed that the Colombian CPV-2a and CPV-2b variants have their common ancestor in sequences in Italy (Figure 3), from which a branch was identified that originated a clade with South American sequences (Uruguay, Argentina and Brazil) derived ca.1990; the Colombian CPV-2a variant is located in this clade. Another branch deriving from the sequences from Italy originated a clade with sequences from Asia (China and India) and Uruguay around 1994, and the CPV-2b variant is located in this clade. 

### 3.4. Structural Analysis

The three-dimensional reconstruction of the VP2 tertiary structure reveals the spatial location of the mutations detected by sequencing for the Colombian CPV-2a and CPV-2b variants. The CPV-2a variant exhibited the Ala297Asn, Tyr324Ile and Ala514Ser mutations, with all mutations being detected in the exposed VP2 regions. The amino acids 297Asn and 324Ile were located in the loop 3, whereas 514Ser was located in a less prominent region. The mutations found in the Colombian CPV-2a sequences are located at the region that divides the barrel and the depression of the 2-fold axis, an important site for the interaction between VP2 and the TfR receptor. The CPV-2b variant exhibited the Phe267Tyr, Tyr324Ile and Thr440Ala mutations. In the three-dimensional VP2 reconstruction, 267Tyr has been found to be located in an unexposed area in the internal structure of the protein. By contrast, 324Ile and 440Ala were detected in more prominent regions of the protein (Appendix A).

### 3.5. Sites under Positive Selection

The positive selection sites were evaluated using the FUBAR and FEL methods. In CPV-2a variant, we found two sites under positive selection: 297 and 324 with a posterior probability of 0.9 and a Bayes factor of 48.8 and 39.4, respectively. In CPV-2b variant we found two positive selection sites 324 and 440 with a posterior probability of 0.9 and a Bayes factor of 39.4 and 53.9, respectively. As previously reported, both variants showed that the 426 amino acid also had a positive selection (Appendix A). By FEL, we also analysed the positive selection between clades and we found that Ala514Ser in the Colombian new CPV-2a had strong positive selection (*p* = 0.033).

## 4. Discussion

Despite the wide distribution of vaccination strategies for CPV-2 and the knowledge obtained from its evolution, CPV-2 remains a viral pathogen that has the greatest of impacts on animal health [20]. In our study, 51.7% of the samples were CPV-2-positive demonstrating that canine parvovirus is the causative agent for most cases of haemorrhagic gastroenteritis in canines [21]. The new CPV-2a variant was detected in 93.1% of the samples, indicating that this new CPV-2a variant previously reported [14] has gradually become the most prevalent virus in Colombia and this site is under positive selection.

Although the youngest animals—aged between 1 and 3 months—represent the population most affected by CPV-2 infection, consistent with the usual presentation of this infection [21], the animal population aged between 7 and 12 months showed a high rate of CPV-2-positive cases. However, animals belonging to the group of mature puppies (aged 7–12 months) showed an incomplete vaccination schedule or absence of any vaccination history (Table 1), rendering them susceptible to CPV-2 because they lack acquired immunity. Only one animal in the sample was aged >12 months; however, this animal had no vaccination history. Despite this unconventional finding, this serves as a starting point to demonstrate the importance and infectious potential of CPV-2 in immunologically mature animals, even in animals that have received a complete vaccination schedule, indicating that the mutational ability of the virus can result in immune response evasion [22]. 

Amino acid sequence analysis revealed important changes in the Colombian CPV-2b (Table 2). The Phe267Tyr change has been reported since the early 2000s in Asian CPV-2b variants [23] and in the Uruguayan CPV-2a antigenic variant [24]. Since its detection, this mutation has consistently appeared in the sequences reported in different studies and has become predominant in the CPV-2 population since 2014, suggesting that this mutation has a positive selection. Although this amino acid was not detected in an exposed area of antigenic change site, the fixation of this mutation reflects an evolutionary advantage for CPV-2 that has not yet been elucidated [6,15].

The Tyr324Ile mutation observed in our CPV-2b variant (Table 2) could be related to the ability of the virus to infect different hosts. The residue 324 is adjacent to the residue 323 and, in combination with the amino acid residue 93 of VP2, these residues reportedly exert effects on the host change of the CPV-2 because they are involved in the binding of the virus with TfR [4,25,26]. This amino acid is found in the loop 3 [15], a moderately exposed VP2 region, which is a part of the ‘shoulder’ of the structure that forms the threefold axis, a site of greater antigenic importance [11]. This mutation was first reported in Asia [27,28], and similar to residue 267, there exists a relationship with Uruguayan variants [24], where this mutation has been reported.

Another mutation observed in our samples regarding the reference variants was Thr440Ala (Table 2). This residue is found in the loop 4 in the most prominent region of the viral capsid—the threefold axis [29]. Therefore, mutations in this region could greatly impact antigenic changes that have implications on the host immune response [24]. Considering that this mutation has undergone a strong positive selection, identifying this mutation in different populations is possible as they underwent an independent evolution [30]. This mutation was first reported in 1993, and it has consistently occurred in the viral populations of CPV-2 since 2005 [15].

It is evident that our findings regarding the Colombian CPV-2b variant are related to the Uruguayan CPV-2a variant. In both cases, these mutations are observed in the same amino acid residues (267, 324 and 440), although they differ in 426. According to the phylogenetic analysis, both the Colombian CPV-2b and Uruguayan CPV-2a variants are present in the same clade as the Asian CPV-2b variants (Figure 2). Based on the study by Grecco et al. [16], the Uruguayan CPV-2a variant originated from the Asian variants, and they named this group the Asia I clade. This clade originated in Asia in the late 80s and arrived in South America between 2009 and 2010, when the dissemination of a divergent CPV-2a variant was reported in Uruguay, where the predominant population at that time was CPV-2c [24].

In the phylogenetic evolutionary analysis (Figure 3), the Colombian CPV-2b variant, although appearing in the same clade as the Uruguayan CPV-2a variant, has a direct relationship with the Chinese variants rather than with the Uruguayan ones. This is possibly attributable to the Uruguayan variants being CPV-2a, whereas in Asia, CPV-2b variants have been reported to have the same mutations as the Colombian CPV-2b sequences [31]. This relationship with the Asiatic variants has been well described recently in Italy, where a case of transcontinental spread of CPV-2c variant was identified with mutation Tyr324Ile and Phe267Tyr characteristically from Asiatic variants [32]. Further, in recent years a local spread of Asiatic-like CPV-2c variant in Italy has been reported [33]. Just as in Uruguay, this spread in Italy demonstrates the intercontinental dissemination of Asiatic CPV-2 variants being able to reach Colombia in a similar way. Structural analysis of CPV-2b has shown that the 440Ala mutation is located in a zone of antibody neutralization. Similarly, it is contiguous to the capsid interaction domain with the TfR cell receptor [34]. 

Regarding the Colombian CPV-2a, an Ala297Asn mutation was shown that has been reported in South American countries, such as Brazil, Uruguay and Argentina [16]. Residue 297 is located at medium exposure zone in the structure of the viral capsid [11] at a site of lower antigenicity and is not located in the most prominent place in the structure [6]. The Colombian CPV-2a-positive and CPV-2b samples contain the Tyr324Ile mutation in the present study, indicating that equivalent changes can occur in different antigenic variants.

The amino acid residue 514 showed the Ala → Ser mutation, which was first reported in Colombia [14] in CPV-2a variants and occurred in 66% of the samples evaluated. However, in our study, 100% of the samples belonging to the new antigenic CPV-2a variant were reported. Surface structure analysis of the virus revealed the spatial position of these mutations. Amino acids 297, 324 and 514 are located in a central region of VP2 (between the depression of the two-fold axis and the canyon). This area is critical for the interaction between the virus and the TfR receptor. The mutations in 297 and 324 are immediately adjacent to the region determined for the coupling of the receptor on the surface of the virus [35]. The mutation in 514 is distant from the virus-receptor interaction zone (Appendix A); however, it is located at the antibody neutralization areas [34]. It is possible that these mutations favour the union between the virus with TfR receptor or may represent a change in the structure of VP2 that allows avoidance of neutralization by antibodies (514Ser) which, added to the fact of being under positive selection, can help explain the reason for this mutation becoming predominant in Antioquia by replacing the viruses that lack these mutations. Interestingly, these same mutations have been reported in Ecuador [36] in a single sample belonging to the CPV-2a antigenic variant, which is closely related to the Colombian CPV-2a variant, highlighting the need to continue the genotypic surveillance of CPV-2 variants circulating in the region due to the possible appearance of new genotypic variants with different possible pathogenic or antigenic potentials.

In agreement with these amino acid variations in the CPV-2a variants of the present study and based on phylogenetic analysis, it is possible that both the Colombian and Ecuadorian CPV-2a samples belong to the clade called South America I. This clade has the peculiarity of containing both South American CPV-2a and CPV-2b variants with the Ala297Asn mutation [16]. The phylogenetic tree presented in Figure 2 shows a well differentiated large clade where all our CPV-2a samples are grouped along with the Ecuadorian CPV-2a. These samples share an identical amino acid sequence, although they differ in some nucleotide level changes that are synonymous variations.

Similarly, in our phylogenetic evolutionary tree construction, Brazilian CPV-2b variants, which contains the Ala297Asn mutation and is present within the clade known as South America I, are most closely related to CPV-2a variants. Although the mutation in 297 groups our CPV-2a variants with variants belonging to the South America I clade, the unique mutations in 324 and 514 differentiate our samples from the others belonging to this clade. This suggests that the Colombian CPV-2 sequences configure a Colombian CPV-2a sub-variant within the CPV-2a variants belonging to the South American clade I, as postulated by Duque-García et al. [14]. 

To understand the evolutionary importance of the mutations found in CPV-2a and CPV-2b variants, an analysis was performed to determine the positive selection sites (Appendix A). Our results confirmed for CPV-2b, that sites 324 and 440 have a strong positive selection, as previously reported in Asian CPV-2 variants [37,38], which are related to the Colombian CPV-2b variant. In the case of CPV-2a, the positive selection sites were 297 and 324. Both sites are adjacent to the virus-receptor interaction domain [34], which demonstrates the importance of these residues in the evolutionary adaptation of the virus and therefore how these mutations have been established in viral populations. Also, by FEL analysis we confirmed that 514Ser in Colombian CPV-2a have strong positive selection, highlighting the importance of our results and the need for active genomic surveillance programs that help to early detect new CPV-2 variants that could be becoming highly prevalent in the region.

According to the phylogenetic analyses performed in the present study, it can be inferred that there are two antigenic variants (CPV-2a and CPV-2b) with different origins currently circulating in Antioquia, Colombia. However, the Colombian CPV-2a variants contain the Tyr324Ile mutation, which is characteristic of the variants belonging to the Asia I clade [16]. To clarify the relationship and origin of the two variants found in Colombia, the phylogenetic evolutionary analysis was performed. Our analysis revealed that two variants have different origins or present specific differential mutations while sharing a common origin.

The phylogenetic evolutionary analysis revealed that the Colombian CPV-2b variant shows a direct relationship with Asian variants that comprise the Asia I clade. This clade arrives in South America, initially in Uruguay in 2009, followed by Colombia in 2012. According to these results, the CPV-2b variant in Antioquia arrived in Colombia in a manner similar to that in Uruguay, i.e., it arrived directly from Asia and not as a migratory process within the continent from Uruguay to Colombia (Figure 3). On the contrary, the Colombian CPV-2a variant was related to variants of the South American clade I, which originated from the European variants that subsequently underwent evolutionary and migratory processes into the interior of the continent to finally configure a clade with characteristics of variants present only in South America [16]. According to the evolutionary analysis, the CPV-2a variant arrived in Colombia between 2005 and 2006.

As previously reported [16], our evolutionary analysis shows that the most ancestral sequences of CPV-2 are the viral sequences reported in the United States at the end of the 1970s. The subsequent arrive of CPV-2 to Europe can be observed, mainly represented by Italian sequences, which give rise to the clade South America-I, in which the Colombian CPV-2a variant is located (Figure 3). Additionally, the Colombian CPV-2b variant iw related to variants of Asian origin, in a similar way to the Uruguayan variants. The evolutionary rate in VP-2 was shown to be similar to that reported by others [16]. However, the resulting trees diverge in their topology without modifying the common ancestry of the clades. These differences in topology in comparison to previously published results may be due to the addition of new and updated sequences to the dataset. It is clear that variations in spatiotemporal sampling added different bias to the analysis, as has been previously reported [39].

Both the CPV-2a and the CPV-2b variant found in our study present the 324Ile mutation (Table 2; Appendix A). This change has been previously reported in cats infected with CPV-2a [38], evidencing that it is an important amino acid in determining host range. Additionally, it has been recently shown that this amino acid residue is close to the virus-receptor interaction domain and that few structural changes are required in CPV-2 to be able to adapt and interact with TfR receptors from different species [34]. This result supports the fact that this mutation has a strong positive selection and is present in all our samples (CPV-2a and CPV-2b). It is possible that other carnivorous species different to canines could be involved in determining the changes in VP2 supporting the adaptation of the virus to carry out an efficient replicative cycle.

## 5. Conclusions

The antigenic variants, CPV-2a and CPV-2b, circulating in Antioquia, Colombia, originated in the South America I clade and Asia I clade, respectively. The mutations detected in CPV-2a variant have gradually undergone positive selection that appears to favour the virus–receptor interaction, rendering this Colombian CPV-2a sub-variant the most predominant in the region. The antigenic implications of the 440Ala mutation in CPV-2b and those related to the virus–receptor interaction should be elucidated in future investigations, with special emphasis on the proximity to sites of virus-receptor and virus-antibody interactions; results that will allow us to understand the functional impact of these genomic changes on the biology, immunology and pathogenesis of CPV-2.

## Figures and Tables

**Figure 1 viruses-12-00500-f001:**
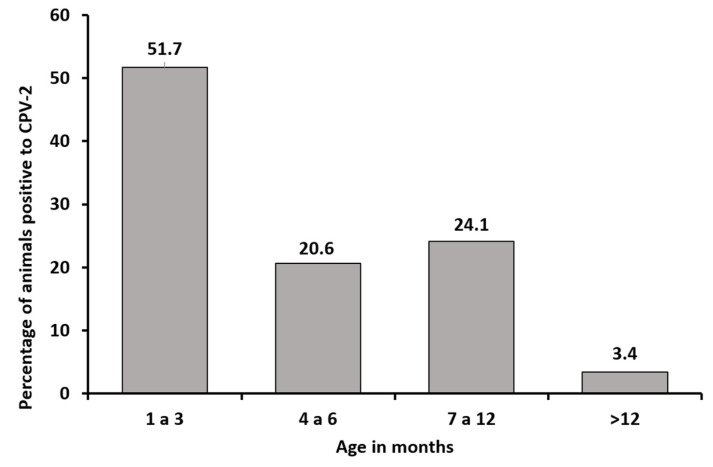
Distribution of CPV-2-positive samples by animal’s age (in months).

**Figure 2 viruses-12-00500-f002:**
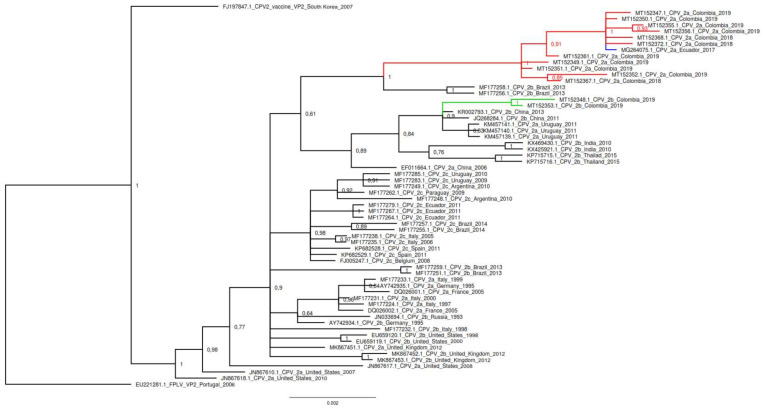
Phylogenetic Bayesian analysis of CPV-2 sequences. The phylogenetic analysis was performed using the nucleotide sequences of VP2 of the Colombian CPV-2a and CPV 2b variants and other sequences belonging to the three antigenic CPV-2a, CPV-2b and CPV-2c variants from different countries, as well as feline panleukopenia virus (FPLV) and original CPV-2 variants. The sequences are identified with the accession number, country, and date of collection. Tree was constructed with Bayes and every clade was supported by Bayesian posterior probabilities (BPP). The tree was rooted with the sequence of FPLV (EU221281.1). Red lines indicate the sequences belonging to the Colombian CPV-2a variant and the blue line indicates that belonging to the Ecuadorian CPV-2a variant related to Colombian samples. Green lines indicate the Colombian CPV-2b variant.

**Figure 3 viruses-12-00500-f003:**
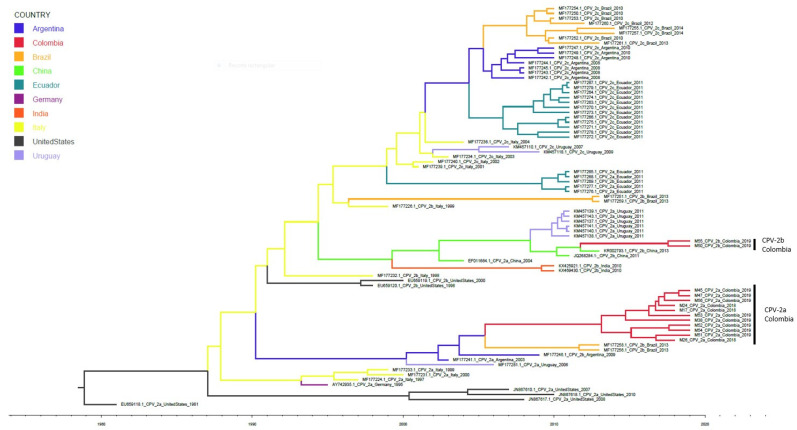
Phylogenetic evolutionary tree of Colombian CPV-2 variants. The tree was generated using CPV-2a, CPV-2b and CPV-2c variants from different countries. The sequences are identified with the accession number, country, and date of collection. The timeline shows the moment of evolutionary divergence of the sequences of each country. The common ancestor for both the Colombian CPV-2a and CPV-2b variants, in red, are sequences of Italian origin, represented in yellow. The Italian variants originated a clade with South American variants in 1990. In 2006, there is a divergence in the Argentine variants that originated the Brazilian variants and the Colombian CPV-2a variant. In 1994, a divergent branch appears that originated the Asian variants. In this clade, there are Chinese and Indian variants. Additionally, Uruguayan sequences related to the Chinese variants are found. In 2012, there is a divergence from the Chinese origin variants that originated the Colombian CPV-2b variant.

**Table 1 viruses-12-00500-t001:** Information on Canine Parvovirus (CPV-2)-positive samples included in the present study.

Sample	Variant	Age	Sex	Race	Vaccination
1	CPV-2a	4 months	Male	Mixed-breed	Without vaccination
5	CPV-2a	3 months	Female	French Bull Dog	Incomplete
7	CPV-2a	1 month	Female	Mixed-breed	Without vaccination
17	CPV-2a	3 months	Female	Golden Retriever	Incomplete
19	CPV-2a	2 months	Female	French Bulldog	Incomplete
20	CPV-2a	2 months	Female	French Bulldog	Incomplete
21	CPV-2a	2 months	Male	French Bulldog	Incomplete
24	CPV-2a	2 months	Female	Beagle	Without vaccination
26	CPV-2a	3 months	Female	Mixed-breed	Without vaccination
29	CPV-2a	2 months	Male	Mixed-breed	Without vaccination
32	CPV-2a	9 months	Female	Pinscher	Without vaccination
33	CPV-2a	2 months	Male	Cocker Spaniel	Incomplete
36	CPV-2a	2 months	Male	Siberian Husky	Without vaccination
37	CPV-2a	7 months	Male	Mixed-breed	Incomplete
38	CPV-2a	6 months	Male	Cocker Spaniel	Complete
40	CPV-2a	4 months	Female	Mixed-breed	Incomplete
41	CPV-2a	4 months	Female	Mixed-breed	Incomplete
43	CPV-2a	9 months	Male	Mixed-breed	Incomplete
44	CPV-2a	3 months	Male	French Bulldog	Incomplete
45	CPV-2a	12 months	Female	Mixed-breed	Without vaccination
47	CPV-2a	7 months	Male	Mixed-breed	Without vaccination
48	CPV-2a	6 months	Male	Mixed-breed	Complete
50	CPV-2b	9 months	Male	Mixed-breed	Incomplete
51	CPV-2a	2 months	Male	Mixed-breed	Without vaccination
52	CPV-2a	2 months	Female	Mixed-breed	Incomplete
53	CPV-2a	4 months	Male	Mixed-breed	Without vaccination
54	CPV-2a	24 months	Male	Mixed-breed	Without vaccination
55	CPV-2b	3 months	Male	German Shepherd	Without vaccination
56	CPV-2a	7 months	Male	Shih Tzu	Incomplete

**Table 2 viruses-12-00500-t002:** Amino acid variations in the samples identified as CPV-2a and CPV-2b in relation to reference variants (in bold).

Sample	Variant	Amino Acid Position					
		267	297	324	426	440	514
**MF177231**	**CPV-2a**	**F**	**A**	**Y**	**N**	**T**	**A**
**EU659119**	**CPV-2b**	**F**	**A**	**Y**	**D**	**T**	**A**
**MF177238**	**CPV-2c**	**F**	**A**	**Y**	**E**	**T**	**A**
1	CPV-2a	F	N	I	N	T	S
5	CPV-2a	F	N	I	N	T	S
7	CPV-2a	F	N	I	N	T	S
17	CPV-2a	F	N	I	N	T	S
19	CPV-2a	F	N	I	N	T	S
20	CPV-2a	F	N	I	N	T	S
21	CPV-2a	F	N	I	N	T	S
24	CPV-2a	F	N	I	N	T	S
26	CPV-2a	F	N	I	N	T	S
29	CPV-2a	F	N	I	N	T	S
32	CPV-2a	F	N	I	N	T	S
33	CPV-2a	F	N	I	N	T	S
36	CPV-2a	F	N	I	N	T	S
37	CPV-2a	F	N	I	N	T	S
38	CPV-2a	F	N	I	N	T	S
40	CPV-2a	F	N	I	N	T	S
41	CPV-2a	F	N	I	N	T	S
43	CPV-2a	F	N	I	N	T	S
44	CPV-2a	F	N	I	N	T	S
45	CPV-2a	F	N	I	N	T	S
47	CPV-2a	F	N	I	N	T	S
48	CPV-2a	F	N	I	N	T	S
50	CPV-2b	Y	A	I	D	A	A
51	CPV-2a	F	N	I	N	T	S
52	CPV-2a	F	N	I	N	T	S
53	CPV-2a	F	N	I	N	T	S
54	CPV-2a	F	N	I	N	T	S
55	CPV-2b	Y	A	I	D	A	A
56	CPV-2a	F	N	I	N	T	S

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
