# Peer review of "Phylogenetic, Evolutionary and Structural Analysis of Canine Parvovirus (CPV-2) Antigenic Variants Circulating in Colombia"

_viruses, 2020, doi:10.3390/v12050500_

Round 1
Reviewer 1 Report
This report on CPV sequences is similar to many others. There is no technical issue with the work apart from the continuing over-emphasis on the residue 426, but they apparently do not wish to change that.
Author Response
We thank the comments. We reduce the emphasis on residue 426 as much as we could in order to include all suggestions of the three different reviewers.
Reviewer 2 Report
The authors have responded to reviewer's comments properly. NJ and Bayesian trees were constructed to confirm the result of ML tree.
However, here is another question concerning about the Bayesian tree (figure 2). What is the meaning of the numbers indicated beside the branch on the figure 2 Bayesian tree? Should not it be posterior probability? They look very strange. Only a clade with posterior probability higher than 0.95 can be considered as monophyletic group. But those numbers are all 0, 0.001, 0.002, and 0.003. Please provide a description of those numbers' meaning to the figure legend.
Author Response
R/. We thank the Reviewer for his/her helpful suggestions, for critical analysis of the manuscript, and for providing new discussion topics. As required a short description was added to the figure legend.
Reviewer 3 Report
This reviewers converns were satisfactorily addressed and the revised manuscript is suitable for publication
Author Response
R/. We thank the Reviewer for his/her helpful suggestions, for critical analysis of the manuscript, and for providing new discussion topics.
This manuscript is a resubmission of an earlier submission. The following is a list of the peer review reports and author responses from that submission.
Round 1
Reviewer 1 Report
A study that reports the sequences of the VP2 genes of CPVs from Columbia. A number of viruses were sequenced that were collected in 2019, and from a variety of infected dogs. After PCR the sequences are determined and analyzed through a series of standard tools, mostly online approaches. The results indicate that the viruses are similar to those in other countries and regions, but with a few mutations specific to this lineage. There is no analysis beyond the sequencing.
The main issue with this is that there is essentially no novelty to this study, and the results and conclusions are similar to the >200 studies that have already been published that look at similar viral sequences in many parts of the world. The point about the viruses being from Columbia does not add any specific interest, as it has been known for a long time that the viruses have a widespread distribution so that similar sequences are found world-wide. There have been active sequencing and analysis programs in at least Brazil and Uruguay in the South American continent, as well as in Italy, China and a few other sites, so that the distribution of the viral analysis is both widespread but not evenly distributed. While this may be useful as a training or surveillance exercise, the amount of new information is very small, so that only an abbreviated presentation is warranted. In the absence of functional analysis there is little evidence that any of the mutations seen are biologically significant.
Some specific points:
1) Line 59 - The focus on residue 426 as being particularly significant has no basis in experimental data. Other references about the antigenic or other structural-associated functions of the virus should be cited.
2) There is no need to show two phylogenies - one would suffice; perhaps that in Fig. 3 is most useful.
3) The structures in Figure 4 can all be show on one footprint; there is certainly no need for two structures.
4) The residues in Table 3 have all been reported many times previously; there is no need to analyses those in this case.
5) The Discussion is 4x as long as it needs to be. A short review of the specific issues around viruses in Columbia may be warranted. Any repetition of the Results section, of the general evolution of these viruses, or the association of mutations that have been reported dozens of times previously can be greatly shortened.
Reviewer 2 Report
In this manuscript, authors performed phylogenetic, evolutionary, and structural analysis of canine parvovirus (CPV-2) antigenic variants circulating in Colombia. Two antigenic CPV-2 variants of two geographically distant origins were found to be circulating in this country. Authors also reported mutations detected in CPV-2a variant have gradually undergone positive selection and speculated that it appears to favour the virus–receptor interaction, rendering this Colombian CPV-2a sub-variant the most predominant in the region.
Overall. this manuscript is well organized and with clear data presentation. Here, I have only a few minor questions and comments as follow:
- Authors constructed the phylogenetic tree with maximum likelihood only. It is advised to use another additional method (e.g. neighbor-joining or Bayesian inference) to construct the phylogeny. The results will be solid if they can be supported by a consistent tree topology obtained from different methods.
- Those CPV-2 sequences obtained in this study should submit to GenBank database and indicate their accession numbers.
- Only several CPV-2 reference sequences out side from South America were included in the phylogenetic analysis. It is advised to include some more reference sequences from other continents.
- There are several methods available for selection pressure estimation in Datamonkey web program, e.g. FEL, SLAC, FUBAR... Why did the authors use FUBAR only?
- The citation in line 374 (Tsao et al., 1991) should list as number of order.
Reviewer 3 Report
In this paper, the authors report the results of a molecular survey for canine parvovirus 2 (CPV-2)in north-western Colombia. The paper adds some novel information to the present knowledge about CPV-2 evolution and genetic variability and can be published after major revision.
Major concerns
1. The authors report that some dogs completed the vaccination protocol at 3 or even 2 months of age. This is not in agreement with the guidelines of the WSAVA and AAHA vaccination groups, postponing the last vaccination to at least 16 weeks of age.
2. Mutation Tyr324Ile has been previously observed in CPV-2c of Asian origin (Mira et al., Transbound Emerg Dis. 2018 Feb;65(1):16-21; Mira et al., Transbound Emerg Dis. 2019 Nov;66(6):2297-2304). This should be added to discussion.
3. Paragraph 2.2. Patient selection and sampling should be better described.
4. Paragraphs 2.5 and 3.2. The maximum likelihood phylogeny should be replaced by a Bayesian approach for more accuracy.
5. Paragraphs 2.7 and 3.4. Structural analysis does not add novel data to what previously published and could be removed.
Minor points
1. Lines 62-63. Add the following reference: Decaro and Buonavoglia, Vet Microbiol. 2012 Dec 10;132(3-4):221-34.
2. lines 317-324. This paragraph should be rewritten according tne the criticism raised at point 1 of Major concerns.